# Neurodevelopmental outcomes in preschool children with congenital heart defects: A case-control study using Ages & Stages Questionnaire

Simone Hansen[1]*, Cathrine Vedel[1], Jesper Steensberg[2], Gorm Greisen[2], Line Rode[3,4], Charlotte Kvist Ekelund[1], Niels Vejlstrup[5], Karin Sundberg[1], Olav Bjørn Petersen[1,4], Ditte S. Jørgensen[1]

1 Center of Fetal Medicine, Department of Gynecology, Fertility, and Obstetrics, Copenhagen University Hospital Rigshospitalet, Copenhagen, Denmark, 2 Department of Pediatrics, Copenhagen University Hospital Rigshospitalet, Copenhagen, Denmark, 3 Department of Clinical Biochemistry, Copenhagen University Hospital - Rigshospitalet, Copenhagen, Denmark, 4 Department of Clinical Medicine, Univerity of Copenhagen, Copenhagen, Denmark, 5 Department of Cardiology, Copenhagen University Hospital Rigshospitalet, Copenhagen, Denmark

* shansensimone@gmail.com

## Abstract

### Objectives

To investigate neurodevelopmental outcome in preschool children with a CHD (congenital heart defect) requiring surgery compared to healthy controls.

### Materials and methods

This study includes all Danish children born between 2016–2018 who underwent surgery within the first year for Ventricular Septal Defect (VSD), atrioventricular septal defect, coarctation of the aorta, double outlet right ventricle, tetralogy of fallot, and Transposition of the Great Arteries (TGA). Exclusion criteria: preterm birth (<37 weeks), twins, and genetic aberrations. Cases were matched with two to four healthy controls on age, sex, gestational age at delivery, and region of birth. Neurodevelopmental outcome at 33–60-months was assessed by the *Ages & Stages Questionnaire* (ASQ). Comparisons were performed by the Mann-Whitney U test and logistic regression analysis and presented as *p*-values and odds ratios (OR) with 95% confidence intervals (CI).

### Results

There were no differences in ASQ score between 105 cases with any CHD (230 (IQR 195–260)) and 179 controls (235 (IQR 205–265)), *p* = 0.12. Cases had five-fold increased odds of a low score compared to controls after adjusting for maternal educational level and children's age at ASQ completion (OR 5.02, 95% CI 1.49;16.90). This was primarily driven by cases with prenatally undetected VSD, where 20%

**Data availability statement:** Yes - all data are fully available without restriction; All relevant data are within the paper and its Supporting information files.

**Funding:** The Project is supported by The Novo Nordisk Foundation grant NNFSA170030576. There was no additional external funding received for this study. We believe the findings will be of interest to a broad readership in pediatric, cardiovascular, and public health fields, and align well with the scope of PLOS ONE.

**Competing interests:** The authors have declared that no competing interests exist.

scored below −2 standard deviations. No differences were found across individual CHD subgroups. In cases with prenatally undetected TGA, the total ASQ score was 185 (IQR 145–190) compared to 220 (IQR 190–270) in prenatally detected TGA, $p = 0.08$.

## Conclusions

The overall ASQ score in children with surgically corrected CHD was comparable to controls. However, cases had five-fold increased odds of a low score, primarily driven by children with prenatally undetected VSD. Additionally, ASQ scores were lower in TGA cases, especially when undetected prenatally, though not significant. These findings suggest follow-up of all children with a CHD requiring surgery including awareness of potential developmental delays.

## Introduction

Congenital heart defects (CHD) are among the most common congenital malformations worldwide with a live-birth incidence of 0.15–3% depending on severity [1–4]. The prognosis varies by the complexity, but survival rates have increased significantly due to advances in prenatal detection and surgical techniques [1–3]. With reduced mortality, the impact of CHD on long-term outcomes have gained a greater focus [5–10]. Increasing evidence shows that children undergoing open-heart surgery during infancy for complex CHD, predominantly investigated in hypoplastic left heart syndrome (HLHS), transposition of the great arteries (TGA), and Tetralogy of Fallot (ToF), are at risk of poorer neurodevelopmental outcomes and at higher risk of some mental health disorders [11–14]. Newer studies have also raised concern of adverse neurodevelopment in children with milder CHD diagnoses [15–18]. However, these studies have had diverging results.

In 2012, The American Heart Association and the American Academy of Pediatrics published a Scientific Statement recommending neurodevelopmental surveillance, screening, and evaluation of all neonates and infants with a CHD requiring open-heart surgery; this statement was subsequently updated in 2024 [8,19]. However, recommendations from other organizations differ from this approach, and The International Society of Ultrasound in Obstetrics and Gynecology emphasizes the importance of further follow-up studies to support current evidence before counseling expectant parents about a possible increased risk of neurodevelopmental delay, except for cases with univentricular circulation [20]. In Denmark, assessment of neurodevelopment is not commonly a part of follow-up programs for children with a CHD requiring surgery.

Thus, we aimed to investigate the neurodevelopmental outcome in Danish preschool children with minor and major CHD requiring surgery within the first year of life compared to healthy controls.

## Materials and methods

This was a case-control study performed at the Center of Fetal Medicine, Department of Gynecology, Fertility, and Obstetrics, Rigshospitalet, Denmark. In Denmark, all women are offered two prenatal ultrasound examinations through a publicly funded healthcare system: A 1st-trimester examination with screening for aneuploidies and a 2nd-trimester examination for assessment of fetal growth and malformations. More than 97% of the pregnant population attend the screening program [21]. Genetic testing is offered in all cases of prenatally detected CHD as up to 20–30% of fetuses with major CHD have a genetic aberration [22,23]. In 2016, the Danish Health Authority centralized pediatric cardiothoracic surgery and catheter intervention at Rigshospitalet, Copenhagen.

This study includes all Danish children born in a three year period (2016–2018) who underwent surgical correction within the first year of life for the most common CHD diagnoses among liveborn children in Denmark: ventricular septal defect (VSD), atrioventricular septal defect (AVSD), coarctation of the aortae (CoA), double outlet right ventricle (DORV), Tetralogy of Fallot (ToF), and transposition of the great arteries (TGA). Cases were identified from patient lists used for surgical quality assurance, and supplemental data were retrieved from The Danish Fetal Medicine Database (DFMD) on January 4, 2021 [24]. The DFMD contains information on all pregnancies including maternal characteristics, fetal growth, pre- and postnatal malformation diagnoses, pre- and postnatal genetic aberrations, delivery, pregnancy and child outcome, and postnatal diagnoses on newborns and infants in Denmark. In the DFMD, prenatal data are retrieved from the prenatal ultrasound system Astraia (Astraia GmbH, Munich, Germany), postnatal data from the National Patient Registry, pregnancy information from the National Birth Registry, and genetic data from the Danish Cytogenetic Central Registry. In Denmark, all residents are assigned a unique central personal registry number at birth or upon immigration, which in the DFMD allows linkage between mother and child. The accuracy of the postnatal CHD diagnoses in the DFMD has previously been evaluated and was correct in 100% of all cases in the greater Copenhagen area between 2015–2018 [2].

All included cases were matched on age, sex, gestational age at delivery, and region of birth with two to four healthy controls retrieved from the DFMD on February 22, 2021. Being healthy was defined as no identified malformations or chromosomal aberrations pre- or postnatally. Furthermore, maternal baseline characteristics were retrieved from the DFMD. Exclusion criteria were preterm birth (<37 weeks), multiple pregnancies, and pre- or postnatally detected genetic aberrations.

The children's neurodevelopmental outcome was assessed by the Danish version of the *Ages & Stages Questionnaire, third edition* (ASQ) [25]. The ASQ is a parent-administered age-specific questionnaire validated against a widely used reference – the Bayley Scales of Infant and Toddler Development – as a developmental screening tool in infants and toddlers with CHD [26]. Each questionnaire consists of six multiple-choice questions in each of the following five areas of development: communication, gross motor skills, fine motor skills, problem-solving, and personal-social functioning. The questions can be answered "yes" (ten points), "sometimes" (five points), and "not yet" (zero points), depending on whether the behavior is present, emerging, or absent. This results in a maximum score of 60 points in each developmental domain and a maximum total score of 300 points. Additionally, ten open-ended questions are included to evaluate the parent´s concerns and the child´s behavior in general.

The ASQ is primarily designed as a screening tool, with items focused on the lower threshold of normal development, which may result in a 'ceiling effect' when children with above average development achieve maximum scores. To reduce the saturation of scores of children with faster-than-normal psychomotor development, 'above-age' questionnaires were used meaning that parents received and completed the questionnaire 3–6 months before the children's questionnaire-specific age. Although this represents a non-conventional use of the ASQ, a similar approach has previously been applied in studies of children born extremely preterm [27,28]. Here it was found to correlate well with Bayley Scales of Infant and Toddler Development and to detect the expected difference between the preterm group and a term born control group.

In this study, questionnaires for 33-, 36-, 42-, 48-, 54- and 60-months old children were used corresponding to the age of approximately 2–5 years. The questionnaires were distributed and completed during the period from March 1, 2021 to

July 31, 2021. Mothers of the included children were sent an ASQ questionnaire and an explanatory letter along with a link to the appropriate ASQ. Parents filled in the questionnaire at home partly from memory and partly from doing the short exercises with their child as described in the questionnaire. To take into account the impact of an inherited genetic component and early interaction, mothers were asked to state the maternal educational level according to a modified version of the UNESCO classification [29]. The score ranged from 1–6 points with completion of 7th-10th grade resulting in 1 point and a PhD resulting in 6 points. Parents were asked to complete the questionnaire within two weeks. In case of non-response, a reminder was sent, and/or phone calls were made.

Based on an assumption that the true prevalence of delayed neurodevelopmental outcome (a developmental score less than −2 standard deviations (SD), i.e., 2.5% in the general population) is 10% in children with surgically corrected CHD, a power calculation based on a non-inferiority design resulted in the need of inclusion of 196 children (98 cases and 98 controls) with a beta = 0.10 og alpha = 0.05 to test the hypothesis that a maximum of 20% (2.5% + 17.5% non-inferiority margin) of the children with CHD had an ASQ score lower than −2 SDs.

Data were analyzed in Stata version 18. Baseline characteristics for cases and controls are presented as frequencies with percentages for categorical variables and means with SDs or medians with interquartile ranges (IQR) for continuous variables depending on the distribution. While the study design was matched, a suboptimal response rate led to the loss of many 'pairs' and therefore the statistical analysis was unmatched. Comparisons were performed by Student's t-test, Mann-Whitney U test, or the Chi$^2$ test. The ASQ scores are presented as medians with IQR and proportions with a low score, defined as a score less than −2 SDs based on the scores from the control group. Comparisons were performed between cases and controls by the Mann-Whitney U test and logistic regression analysis and presented as *p*-values and odds ratios (OR) with 95% confidence intervals (CI). To provide an estimate of the effect size, the score was squared to obtain a symmetric distribution and Cohen's d was calculated. The logistic regression analyses were adjusted for maternal educational level and the child's age when completing the questionnaire. A significance level of <0.05 was chosen.

The study was approved by the Regional Ethics Committee of the Capital Region of Denmark (Journal no.: H-20053569) and the Danish Data Protection Agency (P-2020–755). Written consent was obtained from all participants before the completion of the ASQ.

## Results

In the period 2016–2018, 171 children underwent surgery at Rigshospitalet within the first year of life due to one of the CHD diagnoses included in this study. Cases were screened for eligibility, and 41 children were excluded: twins n = 12, genetic aberrations n = 17, death n = 2, age too old to receive the ASQ n = 5, and no matched controls could be identified n = 5 (Fig 1).

The remaining cases (n = 130) were matched with 504 seemingly healthy controls. Of the controls, 13 were excluded: twins n = 7, genetic aberrations n = 1, death n = 1, and four controls were not able to receive the questionnaire, as their mothers could not receive electronic postage (Fig 2).

Hence, a total of 130 cases and 491 controls were included. The response rate was 80.8% for cases (105/130) and 36.5% for controls (179/491). Table 1 shows baseline characteristics for cases versus controls, whereas baseline characteristics for responders versus non-responders are shown in S1 Table.

The total ASQ scores of cases and controls are shown in Table 2.

We found no significant difference in ASQ score between cases (230, IQR 195–260) and controls (235, IQR 205–265), *p* = 0.12. The Cohen's d was 0.23 (95% CI −0.1 to 0.47). Eleven cases (10.5%) and 4 controls (2.2%) had scores under −2 SDs, yielding a five-fold increased odds of a low score in cases compared to controls that persisted after adjustment for maternal educational level and the children's age at completing the ASQ (OR 5.02 (95% CI 1.49;16.90)). The increased odds of a low ASQ score was mostly driven by cases with surgically corrected VSD, as there was no difference between cases with major CHD and controls (OR 2.15 (0.53;8.74)), while a total of 20% of children with surgically corrected VSD

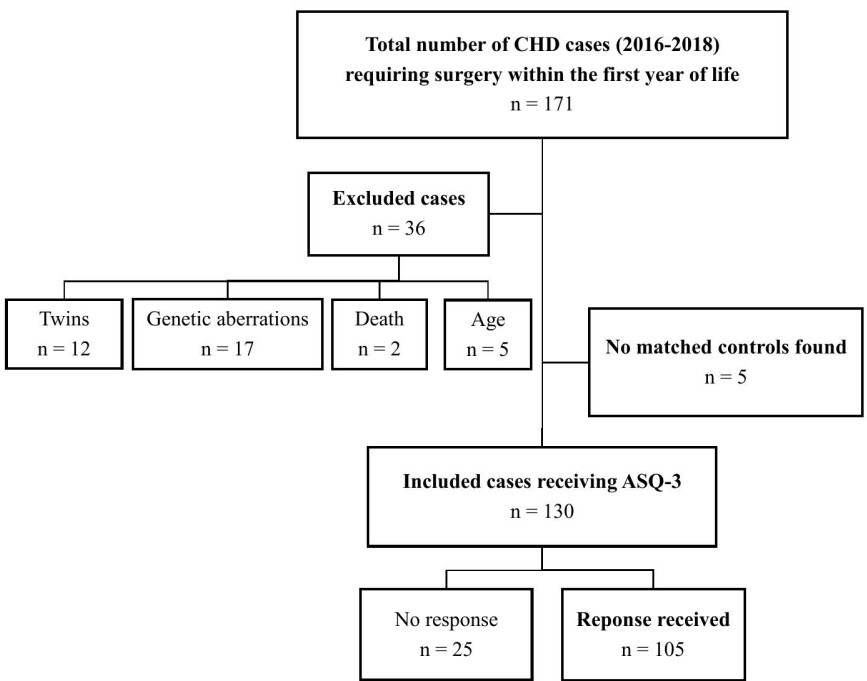

**Fig 1. Flowchart of the study population, cases.** ASQ – Ages & Stages Questionnaire; CHD – congenital heart defect; n – number.

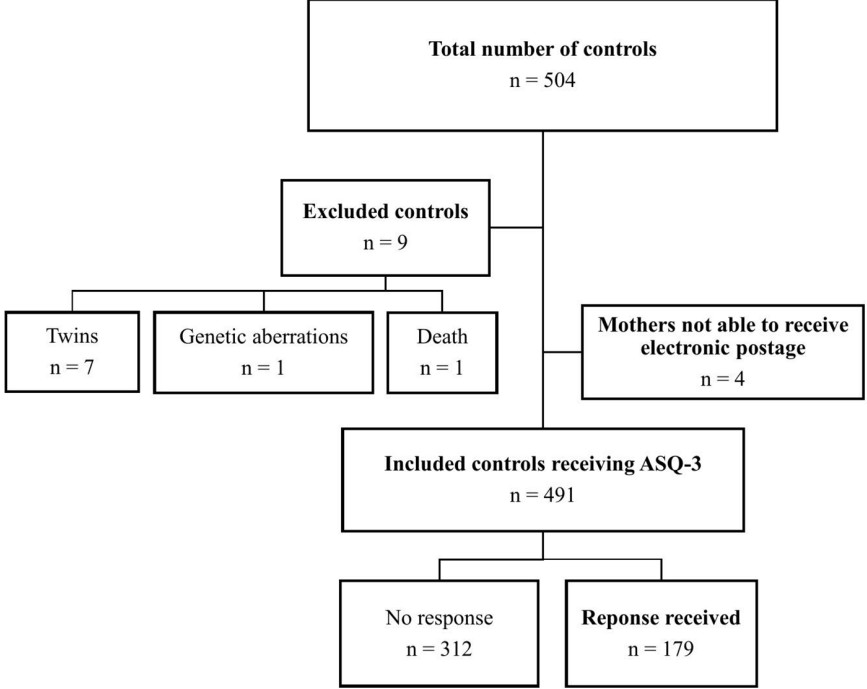

**Fig 2. Flowchart of the study population, controls.** ASQ – Ages & Stages Questionnaire; CHD – congenital heart defect; n – number.

**Table 1. Baseline characteristics for children with surgically corrected CHD and controls.**

| | Cases, n = 105 | Controls, n = 179 | Comparison p-value* |
|---|---|---|---|
| Maternal age, mean (SD) | 30.2 (4.7) | 29.9 (4.9) | 0.62 |
| Maternal BMI, median (IQR) | 23 (21-27) | 22 (20-24) | **0.00** |
| Smoking (yes), n (%) | 4 (3.9) | 11 (6.3) | 0.38 |
| Nulliparous, n (%) | 54 (52.9) | 89 (51.7) | 0.85 |
| Maternal educational level, median (IQR) | 4 (3-5) | 4 (3-5) | 0.68 |
| GA at delivery in days, median (IQR) | 277 (271-281) | 278 (272-284) | 0.13 |
| Birthweight in grams, median (IQR) | 3396 (3055-3710) | 3470 (3164-3850) | **0.04** |

BMI – body mass index; GA – gestational age; IQR – interquartile range; n – numbers; SD – standard deviation.

* Significance level <0.05.

**Table 2. Ages & Stages Questionnaire scores and odds ratios for low score in cases and controls.**

| Diagnose, n | Total ASQ score Median (IQR) | Difference in medians compared to controls p-value* | Proportion with low score (<−2 SD) % | Unadjusted odds ratio for low score (95% CI) | Adjusted§ odds ratio for low score (95% CI) |
|---|---|---|---|---|---|
| *Any CHD, n = 105*<br>*Any control, n = 179* | **230 (195-260)**<br>**235 (205-265)** | **0.12** | **10.5**<br>**2.2** | **5.12 (1.59;16.52)**<br>– | **5.02 (1.49;16.90)**<br>– |
| *Major CHD*<br>Cases = 75<br>Controls = 133 | 235 (200-260)<br>240 (205-265) | 0.55 | 6.7<br>3.0 | 2.30 (0.60;8.86) | 2.15 (0.53;8.74) |
| *Tetralogy of Fallot*<br>Cases = 21<br>Controls = 31 | 240 (205-265)<br>240 (220-265) | 0.45 | 9.5<br>0.0 | – | – |
| *Coarctation of the aorta*<br>Cases = 20<br>Controls = 27 | 238 (223-265)<br>240 (205-260) | 0.93 | 5.0<br>0.0 | – | – |
| *Transposition of the great arteries*<br>Cases = 18<br>Controls = 26 | 203 (185-260)<br>235 (170-265) | 0.53 | 11.1<br>7.7 | – | – |
| *Atrioventricular septal defect*<br>Cases = 9<br>Controls = 17 | 240 (235-250)<br>235 (215-250) | 0.21 | 0.0<br>0.0 | – | – |
| *Double outlet right ventricle*<br>Cases = 7<br>Controls = 12 | 230 (170-265)<br>233 (165-260) | 0.76 | 0.0<br>16.7 | – | – |
| *Ventricular septal defect*<br>Cases = 30<br>Controls = 46 | 218 (170-245)<br>238 (200-270) | 0.06 | 20.0<br>0.0 | – | – |

ASQ – Ages & Stages Questionnaire; CHD – congenital heart defect; CI – confidence interval; IQR – interquartile range; n – number.

* Significance level <0.05.

§Adjusted for maternal educational level and age at completing questionnaire.

had a score <−2 SDs. None of the cases with VSD were prenatally detected or genetically tested. When looking at the total ASQ score in different CHD subtypes, there were no significant differences, see Table 2. In cases with TGA (n = 18) we found an ASQ score of 203 (IQR 185–260) compared to 235 in controls (IQR 170–265), *p* = 0.53. A subanalysis of cases with a prenatally undetected TGA (n = 3) showed an ASQ score of 185 (IQR 145–190) compared to 220 (IQR

190–270) in cases with a prenatally detected TGA (n = 15), *p* = 0.08. See Table 3 for additional ASQ scores of all prenatally detected and undetected CHD subtypes.

## Discussion

In this study, we investigated neurodevelopmental outcomes in Danish preschool children with surgically corrected CHD compared to healthy matched controls using the parent-administered ASQ. The overall differences in ASQ scores between cases and controls were at most moderate, but the odds of a low score (<−2 SDs) in cases was increased five-fold compared to controls. This finding was mainly driven by low ASQ scores in children with a surgically corrected VSD.

Numerous studies have identified elevated risks of poor neurodevelopmental outcomes in children with CHD [5–18]. The comparable ASQ scores observed between cases and controls in this study may reflect several factors, including a well-established prenatal screening program with a high participation rate, a high prenatal detection rate of major CHD, centralized multidisciplinary care with early and timely surgical intervention, and routine genetic testing, which together may influence both case selection and neurodevelopmental outcomes [2,4,21,22]. Notably, univentricular heart (UVH) – the most severe form of CHD, associated with increased risk of poor neurodevelopmental outcomes – was not included, reflecting the very small number of survivors eligible for population-based follow-up in Denmark due to high prenatal dete-cetion (>99%) and frequent pregnancy termination [4,8,20,30]. Furthermore, all included children underwent corrective surgery in accordance with current national guidelines at a centralized tertiary center (Rigshospitalet, Denmark), indicating

**Table 3. Ages & Stages Questionnaire scores in children with prenatally detected and undetected CHD.**

| Diagnose, n (%) | Total ASQ score Median (IQR) | Difference in medians detected vs. undetected *p*-value* | Proportion with low score (<−2 SD) n (%) | Unadjusted odds ratio for low score (95% CI) | Adjusted§ odds ratio for low score (95% CI) |
|---|---|---|---|---|---|
| *Any CHD, n = 105* **Detected = 53 (50.5) Undetected = 52 (49.5)** | **235 (200-260) 225 (185-255)** | **0.28** | **4 (7.6) 7 (13.5)** | **1.91 (0.52;6.95)** | **1.85 (0.41;8.32)** |
| *Major CHD* Detected = 53 (70.7) Undetected = 22 (29.3) | 235 (200-260) 228 (195-265) | 0.90 | 4 (7.6) 1 (4.6) | 0.58 (0.06;5.54) | 0.48 (0.04;5.37) |
| *Tetralogy of Fallot* Detected = 16 (76.2) Undetected = 5 (23.8) | 240 (200-263) 210 (205-265) | 0.92 | 2 (12.5) 0 (0.0) | – | – |
| *Coarctation of the aorta* Detected = 10 (50.0) Undetected = 10 (50.0) | 238 (220-245) 235 (225-270) | 0.75 | 1 (10.0) 0 (0.0) | – | – |
| *Transposition of the great arteries* Detected = 15 (83.3) Undetected = 3 (16.7) | 220 (190-270) 185 (145-190) | 0.08 | 1 (6.7) 1 (33.3) | – | – |
| *Atrioventricular septal defect* Detected = 6 (66.6) Undetected = 3 (33.3) | 240 (235-240) 250 (235-285) | 0.41 | 0 (0.0) 0 (0.0) | – | – |
| *Double outlet right ventricle* Detected = 6 (85.7) Undetected = 1 (14.3) | 240 (200-265) 170 (-) | 0.57 | 0 (0.0) 0 (0.0) | – | – |
| *Ventricular septal defect* Detected = 0 (0.0) Undetected = 30 (100.0) | – 218 (170-245) | – | – 6 (20.0) | – | – |

ASQ – Ages & Stages Questionnaire; CHD – congenital heart defect; CI – confidence interval; IQR – interquartile range; n – number.

* Significance level <0.05.

a consistently high standard of surgical care. This is supported by a nationwide study from 2018–2022 reporting an overall postnatal mortality of only 9.8% (32/326) among children with major congenital heart disease, approximately half of whom were not considered for or scheduled for surgery [4]. These factors may collectively help explain the comparable ASQ scores observed between cases and controls.

Although our data did not show any overall differences in ASQ scores between cases and controls, an increased odds of a low ASQ score in children with a surgically corrected VSD was found. Previous studies investigating neurodevelopmental outcomes in children with surgically corrected VSD have shown inconsistent results: Several studies have reported neurodevelopmental challenges either in cohorts exclusively consisting of children with VSD repair in infancy or in studies including both VSD and other major CHD diagnoses, with Billotte et al. also using the ASQ [12,15,17]. In contrast, other studies examining the long-term consequences of children with surgically corrected VSD have not found evidence of neurodevelopmental delays [31,32]. However, it is important to note that all of the above-mentioned studies exhibit methodological differences, including variations in sample sizes, diagnostic or screening tools, and the ages at which assessments were conducted, which complicates direct comparison of the findings.

In contrast to other CHD subtypes in this study, none of the VSDs were detected prenatally and, consequently, the parent's had not been offered genetic testing prenatally. However, genetic tests are also performed postnatally when genetic syndromes are suspected. Furthermore, the VSD subgroup was not differentiated into muscular and perimembranous defects. Taking existing literature into account, it may be considered that an undiagnosed genetic component could have contributed to the neurodevelopmental challenges observed in this group: Previous studies have demonstrated that isolated perimembranous VSDs are more likely to require postnatal intervention due to their anatomical characteristics. Furthermore, perimembranous VSDs are more frequently associated with chromosomal abnormalities [33,34]. Hence, it seems reasonable to consider a contribution from a genetic component to the low ASQ score in some of the VSD cases in this study. This speculation aligns well with the findings of Kaltman et al., who reported neurodevelopmental outcomes within normal limits in infants undergoing surgical VSD repair, although worse outcomes were observed in the presence of a suspected or confirmed genetic condition [32]. Alternatively, the lack of prenatal detection may have delayed postnatal management, hence indirectly impacting the neurodevelopment. The finding indicates that attention to neurodevelopmental outcomes in children with a surgically corrected VSD may be considered, especially when prenatally undetected. This consideration is consistent with the conclusion by Billotte et al. as well as the American Heart Association and the American Academy of Pediatrics, who suggest implementation of formal neurodevelopmental evaluation in the CHD population, regardless of severity [15,19].

When analyzing the different CHD subtypes, we observed a lower ASQ score in children with TGA, particularly in prenatally undetected cases. Prior research have demonstrated that TGA is associated with an elevated risk of neurodevelopmental delays [35,36]. The small sample size in this TGA subgroup might explain why the data did not reach statistical significance. The observation of the lowest total ASQ score in cases with prenatally undetected TGA may be explained by postnatal circulatory compromise or suboptimal management prior to surgery, a speculation supported by existing literature: Peyvandi et al. found that prenatal diagnosis of TGA decreases risk of preoperative brain injury and is associated with a faster rate of postnatal brain growth [37]. Moreover, early surgical intervention, facilitated by prenatal detection, may prevent further brain injury by minimizing postnatal exposure to abnormal cerebral perfusion [38]. These findings underscore the importance of prenatal diagnosis in improving neurocognitive outcomes for children with TGA. Furthermore, it reinforces the previously mentioned suggestion of emphasized focus on neurodevelopmental outcome in children with prenatally undetected CHD.

This study has several strengths. The nationwide inclusion of cases and controls from all regions of Denmark and the high response rate among cases makes the results representative for the general population. Cases were matched with their respective controls at multiple parameters ensuring a high degree of comparability. Whereas we abstained from paired statistical analysis due to suboptimal response rates, reasonable balance of potential confounders were obtained.

Furthermore, it was possible to retrieve additional data on pregnancy and delivery outcomes from all included cases and controls, ensuring a comprehensive dataset showing no essential differences in baseline characteristics between the groups.

A limitation of the study is the smaller sample size when analyzing the CHD subgroups, and thus our findings should ideally be validated in a larger study. Nevertheless, the data amount is comparable to other published studies within this field, which is a research area still characterized by sparse evidence. Another limitation concerns the lower response rate among controls compared to cases, which may limit the representativeness of the control group for the general population. As this difference in response rate was anticipated, all cases were matched with two to four controls to strengthen the representativeness of the dataset. Nevertheless, the lower response rate among controls may have led to an overestimation of ASQ scores in this group if parents with more resources or well-performing children have been more likely to respond. However, the observed effect size for the primary outcome was small (Cohen's d = 0.23), suggesting that any such bias is unlikely to have had a major impact on the results. In addition, no clinically relevant differences in baseline characteristics were identified between responders and non-responders, further reducing the risk of selection bias.

Finally, developmental tests in young childhood has only moderate ability to predict cognitive function as measured later in childhood by formal intelligence tests [39]. This study only represents the preschool period in the children's development with no formal test of cognitive function or academic achievements. In the future, with consecutive follow-up data, it will be possible to investigate the long-term neurodevelopmental consequences in these children with surgically corrected CHD.

## Conclusions

This nationwide study of a population with universal prenatal screening for CHD found that the overall ASQ score in preschool children with surgically corrected CHD was comparable to controls. However, cases had a five-fold increased odss of a low score, primarily driven by children with a prenatally undetected VSD. Additionally, we observed a lower, however not significant, ASQ score in cases with TGA, especially when not detected prenatally. These findings suggest that the follow-up of all children with a CHD requiring surgery should include awareness of potential developmental delays.

## Supporting information

**S1 Table. Baseline characteristics for responders and non-responders.** SD – standard deviation; BMI – body mass index; IQR – interquartile range; n – numbers; GA – gestational age. *Significance level <0.05.
(JPG)

## Author contributions

**Conceptualization:** Simone Hansen, Cathrine Vedel, Jesper Steensberg, Gorm Greisen, Line Rode, Charlotte Kvist Ekelund, Niels Vejlstrup, Karin Sundberg, Olav Bjørn Petersen, Ditte S. Jørgensen.

**Data curation:** Simone Hansen, Cathrine Vedel, Jesper Steensberg, Olav Bjørn Petersen, Ditte S. Jørgensen.

**Formal analysis:** Cathrine Vedel, Gorm Greisen, Line Rode.

**Funding acquisition:** Olav Bjørn Petersen.

**Investigation:** Simone Hansen, Cathrine Vedel, Ditte S. Jørgensen.

**Methodology:** Simone Hansen, Cathrine Vedel, Jesper Steensberg, Gorm Greisen, Line Rode, Charlotte Kvist Ekelund, Olav Bjørn Petersen, Ditte S. Jørgensen.

**Project administration:** Simone Hansen, Cathrine Vedel, Ditte S. Jørgensen.

**Resources:** Olav Bjørn Petersen.

**Software:** Cathrine Vedel.

**Supervision:** Cathrine Vedel, Jesper Steensberg, Gorm Greisen, Line Rode, Charlotte Kvist Ekelund, Niels Vejlstrup, Karin Sundberg, Olav Bjørn Petersen, Ditte S. Jørgensen.

**Validation:** Simone Hansen, Cathrine Vedel, Jesper Steensberg, Gorm Greisen, Line Rode, Charlotte Kvist Ekelund, Niels Vejlstrup, Karin Sundberg, Olav Bjørn Petersen, Ditte S. Jørgensen.

**Visualization:** Simone Hansen, Cathrine Vedel, Gorm Greisen, Ditte S. Jørgensen.

**Writing – original draft:** Simone Hansen, Cathrine Vedel, Ditte S. Jørgensen.

**Writing – review & editing:** Simone Hansen, Cathrine Vedel, Jesper Steensberg, Gorm Greisen, Line Rode, Charlotte Kvist Ekelund, Niels Vejlstrup, Karin Sundberg, Olav Bjørn Petersen, Ditte S. Jørgensen.

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
