## [Decision Letter · Decision Letter 0]

13 Nov 2025

Dear Dr. Hansen,

Thank you for submitting your manuscript to PLOS ONE. After careful consideration, we feel that it has merit but does not fully meet PLOS ONE’s publication criteria as it currently stands. Therefore, we invite you to submit a revised version of the manuscript that addresses the points raised during the review process.

We look forward to receiving your revised manuscript.

Kind regards,

Hany Mahmoud Abo-Haded, MD

Academic Editor

PLOS ONE

Journal Requirements:

2. Please update your submission to use the PLOS LaTeX template. The template and more information on our requirements for LaTeX submissions can be found at http://journals.plos.org/plosone/s/latex .

[The Project is supported by The Novo Nordisk Foundation grant NNFSA170030576].

4. Please note that your Data Availability Statement is currently missing the accession number of each dataset OR a direct link to access each database. If your manuscript is accepted for publication, you will be asked to provide these details on a very short timeline. We therefore suggest that you provide this information now, though we will not hold up the peer review process if you are unable.

6. We notice that your supplementary table is included in the manuscript file. Please remove them and upload them with the file type “Supporting Information”. Please ensure that each Supporting Information file has a legend listed in the manuscript after the references list.

Reviewers' comments:

Reviewer's Responses to Questions

**Comments to the Author**

1. Is the manuscript technically sound, and do the data support the conclusions?

Reviewer #1: Yes

Reviewer #2: Yes

2. Has the statistical analysis been performed appropriately and rigorously?

Reviewer #1: Yes

Reviewer #2: Yes

3. Have the authors made all data underlying the findings in their manuscript fully available?

Reviewer #1: Yes

Reviewer #2: Yes

4. Is the manuscript presented in an intelligible fashion and written in standard English?

Reviewer #1: Yes

Reviewer #2: Yes

Reviewer #1: The manuscript addresses an important and clinically relevant topic: neurodevelopmental outcomes in preschool children with surgically corrected congenital heart defects (CHD), assessed with the Ages & Stages Questionnaire (ASQ). The study is well-designed, based on nationwide data, and uses appropriate methodology. My comments are as following:

The exclusion criteria may have inadvertently omitted critical cases, particularly those admitted to the neonatal ICU. Moreover, the outcomes may also depend on the surgical success rate; please provide more detailed information on this aspect.

The methodology related to the use of the ASQ requires further clarification and justification, as the current description is not sufficient.

The follow-up period is relatively short, which limits the ability to assess long-term neurodevelopmental outcomes.

Reviewer #2: Congratulations on the good job.

1. Abstract

i. Excellent summary

ii. Proper inference

2. Framing – disclosures, funding and ethical issues

a. Excellent

3. Introduction, methods, results and conclusion

a. The response rate amongst controls is 36.5%. Such may mean skewed result. Kindly relook the impact on the results.

b. Discussion section shouldn’t take a lot of explanations to convince readers the methodology was proper. You better remove it and can drop some points at the methods section if you wish.

c. Prevalence of congenital heart diseases and screening isn’t a target, and doesn’t need discussion. Was better finalized on the introduction. It took a content here.

d. The subgroup analysis had created minimal sample sizes, which remained to be a limitation.

e. Long – term neurodevelopmental outcomes

f. Term of astonishment like surprisingly better is avoided

**Do you want your identity to be public for this peer review?** For information about this choice, including consent withdrawal, please see our Privacy Policy

Reviewer #1: No

Reviewer #2: **Yes:** Abraha Gebreegziabher Hailu, MD

---

## [Author Response · Author response to Decision Letter 1]

26 Dec 2025

Dear Editor and reviewers,

We would like to thank you for your comments and suggestions aimed at improving our manuscript, “Neurodevelopmental Outcomes in Preschool Children with Congenital Heart Defects: A Case-Control Study using Ages&Stages Questionnaire”.

We sincerely appreciate the time and effort invested providing constructive feedback. We have carefully addressed all points raised. Please do not hesitate to contact us should any clarifications or further revisions be required.

We hope you will find the revised manuscript acceptable for PLOS ONE.

Sincerely and on behalf of the co-authors,

Simone Hansen, MD

---

## [Decision Letter · Decision Letter 1]

4 Jan 2026

Neurodevelopmental Outcomes in Preschool Children with Congenital Heart Defects: A Case-Control Study using Ages&Stages Questionnaire

PONE-D-25-33159R1

Dear Dr. Hansen,

We’re pleased to inform you that your manuscript has been judged scientifically suitable for publication and will be formally accepted for publication once it meets all outstanding technical requirements.

Kind regards,

Hany Mahmoud Abo-Haded, MD

Academic Editor

PLOS One

Additional Editor Comments (optional):

Reviewers' comments:

Reviewer's Responses to Questions

**Comments to the Author**

Reviewer #1: All comments have been addressed

Reviewer #2: All comments have been addressed

2. Is the manuscript technically sound, and do the data support the conclusions?

Reviewer #1: Yes

Reviewer #2: Yes

3. Has the statistical analysis been performed appropriately and rigorously?

Reviewer #1: Yes

Reviewer #2: Yes

4. Have the authors made all data underlying the findings in their manuscript fully available?

Reviewer #1: Yes

Reviewer #2: Yes

5. Is the manuscript presented in an intelligible fashion and written in standard English?

Reviewer #1: Yes

Reviewer #2: Yes

Reviewer #1: As a peer reviewer, I have carefully evaluated the submitted manuscript for its scientific validity, methodology, data analysis, and clarity of presentation. The study addresses a clinically relevant topic and is based on appropriate study design and analytical methods. The authors have adequately addressed all comments raised during the review process and have revised the manuscript accordingly. After thorough reassessment of the revised version, I confirm that the manuscript meets the required academic and ethical standards and has successfully passed the peer-review process.

Reviewer #2: (No Response)

**Do you want your identity to be public for this peer review?** For information about this choice, including consent withdrawal, please see our Privacy Policy

Reviewer #1: No

Reviewer #2: No

---

## [Editor Report · Acceptance letter]

PONE-D-25-33159R1

PLOS One

Dear Dr. Hansen,

I'm pleased to inform you that your manuscript has been deemed suitable for publication in PLOS One. Congratulations! Your manuscript is now being handed over to our production team.

Kind regards,

on behalf of

Professor Hany Mahmoud Abo-Haded

Academic Editor

PLOS One